# Conceptual Design of the “Private Car” Self-Isolation Ecosystem for the 2019-nCoV Infection

**DOI:** 10.3390/ijerph191610385

**Published:** 2022-08-20

**Authors:** Yudong Wang, Lanting Wang, Xinggui Wu, Ziyi Ding, Wanbo Zheng, Xingxing Liang, Huaming An

**Affiliations:** Faculty of Public Security and Emergency Management, Kunming University of Science and Technology, Kunming 650000, China

**Keywords:** COVID-19, self-isolation, isolation method

## Abstract

Since the beginning of the COVID-19 outbreak, confirmed and suspected cases of the disease have been increasing rapidly. The isolation of cases is one of the most effective methods for the control and containment of COVID-19 and has been rapidly popularized. Problems with isolation have gradually emerged, such as the inadequate allocation of isolation resources and the failure to properly resettle many of the suspected cases of the 2019-nCoV infection. In this paper, a self-isolation ecosystem of a rapid-deploying negative-pressurized “private car” is proposed for housing patients with 2019-nCoV infection, which could be lightweight, moderately sized and transparent to enable group supervision and communication. This “private car” isolation method aims to achieve self-isolation of patients and essentially solves the problem of where and how to isolate suspected cases while saving isolation resources and preventing the large-scale transmission of COVID-19.

## 1. Introduction

Since the outbreak at the end of 2019, the epidemic of novel coronavirus pneumonia (hereinafter referred to as “COVID-19”) has been rampant throughout the world. As of 30 July 2022, Beijing time, more than 575.95 million confirmed cases have been reported globally, and more than 6.39 million of those infected have died. The speed of transmission, scope of infection and difficulty of prevention and control in COVID-19 are far greater than those of AIDS, Ebola severe acute respiratory syndrome, Middle East respiratory syndrome, avian influenza and other new infectious diseases [1]. Presently, the isolation policy has been introduced as an effective measure to limit the spread of COVID-19, reducing the number of infected people.

According to the practice of different countries in the prevention and control of the COVID-19 epidemic, this study analyzes in depth the advantages and disadvantages of various mobile medical platforms. These include field tent hospitals, field shelter hospitals and vehicle-mounted mobile hospitals for the screening, isolation, transportation, diagnosis and treatment of sudden major infectious diseases. The most common types of mobile medical platforms mainly include field tent hospitals, field shelter hospitals, vehicle-mounted mobile hospitals, hospital ships, medical rescue helicopters, etc. Field tent hospitals, field shelter hospitals and vehicle-mounted mobile hospitals are the main forms of ground medical rescue, a hospital ship is the carrier of maritime medical rescue, and a medical rescue helicopter is an important carrier of air medical rescue. A field tent hospital is convenient for delivery and rapid deployment. Equipped with corresponding technical support equipment, it can meet different needs, such as inspection, screening, treatment and isolation. However, it is limited in use when encountering extreme weather conditions, such as storms [2]. The operation environment of the field shelter hospital is stable and good, but the delivery process depends on large vehicles. When the road is damaged or interrupted by geological disasters, it is difficult to enter the support area in time. Vehicle-mounted mobile hospitals, hospital ships and medical rescue helicopters are mostly used as transport vehicles for the sick and wounded, and have certain rescue equipment and conditions for their use. A vehicle-mounted mobile hospital has a strong flexibility and can shuttle between multiple areas, but its carrying capacity is limited. A hospital ship has the largest carrying capacity and is suitable for sea rescue or the transfer of several patients [3]. A medical rescue helicopter is fast, but it has take-off and landing conditions and requirements and can only carry a small number of patients at a time.

As one of the most effective means of controlling the COVID-19 epidemic, isolation plays an important role in relieving the risk of transmission. The COVID-19 outbreak isolation strategy detailed here was used to isolate, diagnose, and treat three levels of infection: the critically ill patients in Huoshenshan Hospital, the mildly ill patients in a module hospital, and the suspected cases. However, due to the lack of isolation resources in the severely affected areas, many suspected cases could not be properly resettled. Lack of proper isolation of patients with confirmed or suspected COVID-19 has a great potential for enabling the spread of infection and has a substantial impact on the prevention and control of the epidemic.

As the number of confirmed and suspected cases of COVID-19 increases rapidly with the continuous development of the epidemic, “self-isolation” is believed to be the best way to isolate suspected cases that have no designated place to be resettled. Efforts to expand the scale of self-isolation and improve the isolation environment slow down the development of the epidemic and provide people in epidemic areas with the necessary protection [4].

Self-isolation has a direct impact on the development of COVID-19. One of the important ways to solve the shortage of isolation resources is to constantly innovate ways for patients to self-isolate. This paper describes the environmental, spatial, disinfection-related, diagnostic, and treatment-design aspects of a proposed “private car” self-isolation ecosystem, as well as the precautions to consider during the development of this system. This system was primarily designed to control the transmission of 2019-nCoV, reduce the hidden danger of 2019-nCoV infection, fully meet the needs of patients, and save isolation resources to provide a novel conceptual isolation strategy for future responses to the massive spread of infectious diseases [5,6,7,8].

## 2. Development and Design of the System

### 2.1. Design Brief of the System

Under the current emergency response, in the large indoor space equipped with ventilation and air conditioning, low-cost and low-technology are used to provide an observation space with negative pressure laminar flow and certain air density in a rapid and large-scale manner, which is suitable for individuals with a self-care ability to have a long isolation period. At the same time, keeping the line of sight unobstructed significantly increases the efficiency of centralized care and maintains the possibility of doctor–patient and patient–patient communication. The applicable environment for the system is an outdoor/field environment.

### 2.2. Development Principles of the System

Certain principles must be applied when developing a self-isolation system to ensure the resulting system is convenient and standardized as follows:(1)Applicability: The development of the system follows the principle of user priority because the function of the system is to serve and meet the needs of users.(2)Scalability: Scalability is a measure of the quality of a system. If a system only solves the existing functions and cannot adapt to the new functions in time, when the users of the system have other new requirements in the process, the system needs to be redesigned. This would involve a waste of human and material resources and affects the user experience, so good scalability is beneficial to both users and developers.(3)Reusability: The content of many modules in a system is developed in advance so that many similar tasks can be realized through reuse, which not only improves efficiency but also greatly enhances usability.

### 2.3. Design of the Isolation Environment of the System

According to “Ending Isolation and Precautions for People with COVID-19: Interim Guidance” of the WHO, the negative pressure isolation unit can be formed by using the existing 3 M (10 ft) outdoor etiquette tent skeleton and some PVC coated fiber cloth skin from the market, replacing one or more local transparent PVC films, making proper air tightness enhancement, adding necessary air inlets and outlets, and taking material reinforcement measures. No matter whether there is a fresh air supply or not, the isolation unit of the infection department needs a special air distribution; therefore, the air flow from the top to the bottom, or from the height of the medical staff to the head of the hospital bed, and the occurrence of turbulence are minimized [9]. Therefore, the best way is to undertake laminar flow organization. For this reason, the author considered a special low-cost laminar flow air duct and air outlet in the subsequent optimization. If the cost is limited and the laminar flow measures are even to be omitted, the simplest method is to set the air inlet and outlet along the diagonal of the space, so that the turbulence range will be slightly smaller than the direct arrangement. The isolation units that complete the above tasks can be used alone, but they should be arranged in groups to form an alternate arrangement of nursing corridors and pipeline corridors [10]. The intensive arrangement of multiple units is conducive to the efficient use of positive and negative pressure ventilation systems and the intensive arrangement of medical pipelines. A flexible airtight valve is arranged at the back of a tent to connect medical pipes, such as oxygen, sputum suction, negative pressure pipes and data power supply (it can also be assumed that this equipment, both gas and liquid, uses mobile equipment under limited conditions). The specific plane of the two types of systems must be in accordance with the plane of the specific venues as the external environment [11].

### 2.4. Design of the Isolation Space of the System

#### 2.4.1. Design Principles for the Isolated Space of the System

The isolation space, as the main body of the isolation system, must meet the following requirements:(1)In principle, only one patient is allowed in each isolated space.(2)Usually, people will be infected with COVID-19 by droplets only when they are in close contact with the source of infection. Because the spray distance for droplets is only about 1 m without the help of external conditions, it is relatively safe to be 1 m away from the source of infection. At the same time, the distance between two isolation units in the isolation area should be >1 m.(3)When the isolation conditions are limited, the quarantined personnel who are most susceptible to infection should be placed first, and the quarantined personnel infected with the same pathogen should be placed in the same isolation space [12].

#### 2.4.2. Design of the Isolation Space of the System

A single, isolated space must be air-tight; have a separate and continuous-operation air conditioning system; be equipped to meet the clothing, food, shelter, transportation, disinfection, and treatment needs of isolated patients; and must be readily adjustable as the epidemic situation changes. Private cars as isolation spaces meet the above requirements.

Due to the limited internal space of a private car, the equipment used by the isolated patient may need to be temporarily assembled or converted from other functions. For example, the table and chairs in the car need to be unfolded at night as part of the bed panel. The patient bed must be ergonomically designed for isolation. According to specifications for human body size and recreational vehicle industry standards, designers must ensure the minimum functional dimension of a person’s moving space as the minimum width of the bed; that is, it must be ≥700 mm. The net length of the bed surface for adults should be ≥1980 mm, and the minimum height of the bedplate and the space above should be ≥800 mm [13]. Designers must meet most of the living needs of the isolated people by maximizing folding functions for furniture or fixtures and enabling convenient storage to save space in the isolated private car. At the same time, the isolation space must have a living space that is separate from the contaminated space and can be disinfected according to the disinfection requirements for each space. This design improves the comfort of the patients’ living environment.

A private car used for isolation differs from an ordinary vehicle. The power generated by a self-isolation private car must meet the daily needs of an isolated patient while maintaining the capability to be driven. Acquiring or supplementing the electric energy generated in a self-isolation private car is one problem in the process of private car transformation. The electric energy generated by private cars typically is only enough to support their basic functions and their main source of energy is gasoline and diesel. In the process of designing the private car as an isolated space, the space and weight of a generator are prohibitive, so designers must consider the use of an additional, solar-charged vehicle battery as the main storage device of electric energy.

### 2.5. Design of Isolation and Disinfection of the System

#### 2.5.1. Principles of System Isolation and Disinfection

The principles of system disinfection were designed with the consideration of the potential for novel coronavirus resistance and according to the General Principles of Disinfection at the Source. Contaminants and contaminated environments within the isolated environment must be accessible for disinfection at any time, and patients in isolation areas are subject to terminal disinfection after transfer [14]. Disinfection focuses on living items, diagnostic equipment, and isolated air that may be contaminated by patients.

The hygiene standards for performing disinfection at any time states that no pathogens or hemolytic streptococci must be detectable after disinfection in respiratory tract infections (indirect pollution index); ≥90% of natural bacteria are killed in the air after disinfection, and pathogenic microorganisms or indirect indicator bacteria cannot be used as a criterion to evaluate the killing effect. For the disease to be evaluated, disinfection should be carried out 1–2 times a day. Pathogenic microorganisms in the source of the epidemic area must not be detected, and pathogenic microorganisms must not be detected after disinfection of excretion or secretions [15].

#### 2.5.2. Selection of Disinfectants and Methods for Disinfection of the System

We applied the following principles of disinfectant selection: Firstly, use disinfectants without toxicity, residue, corrosive properties, and risk of other secondary contamination. Secondly, use disinfectants with a fast germicidal speed, high germicidal efficiency, wide popularity, and minimal environmental impact. Most importantly, use the least expensive disinfectants at the lowest possible concentrations. Finally use disinfectants that are simple and convenient to use.

According to one study, COVID-19 is sensitive to physical and chemical factors, has a moderate resistance to heat and can be effectively killed after 30 min of ultraviolet irradiation. 2019-nCoV can be effectively killed by chlorine-containing disinfectant substances within a defined period of time. Chlorinated disinfectants destroy 2019-nCoV by oxidizing and denaturing proteins and other substances in the virus.

The environment and articles in the isolated space must be disinfected by keeping windows open for ventilation and using chlorine-atomizing or alcohol-atomizing disinfection equipment. Such equipment atomizes the disinfectant into hundreds of millions of nanometer-sized, ultrafine particles, which more effectively kill the germs in the air and on objects. At the same time, antibacterial particles can also effectively penetrate places that other disinfection equipment cannot reach in the transformed isolation space, such as the air-conditioning system, under the dashboard, under the carpet, and within the roof interlayer of the private car.

#### 2.5.3. Evaluation of the Isolation and Disinfection Effects of the System

In the GB15981-1995 Evaluation Method and Standard of Disinfection and Sterilization Effect, there are two methods for determining the disinfection effect of the self-isolation system: the qualitative disinfection experiment and the quantitative disinfection experiment. The qualitative disinfection experiment is used to determine the presence of bacteria in the sterilized samples and the preliminary identification of the sterilization effect. The quantitative disinfection experiment is used to determine the number of residual microorganisms in the sterilized samples and is used to further evaluate the efficacy of sterilization.

The specific method for evaluating the disinfection effect combines the reverse transcription polymerase chain reaction (RT-PCR) with the cell culture. RT-PCR is a method using fluorescent groups in the PCR reaction system, fluorescence signal accumulation to monitor the whole PCR process in real time [16,17] and, finally, a standard curve to analyze the unknown template quantitatively. Although this method may be used to efficiently and quickly detect nucleic acid samples and confirm the existence of 2019-nCoV nucleic acid, it cannot be used to analyze the nature of the 2019-nCoV in the sample. Therefore, a second method of cell culturing has been proposed. The cell culture method is one of the traditional 2019-nCoV detection methods and can differentiate between the 2019-nCoV strains. This method can be used to study pathogens. Its disadvantage is that it takes a long time and is not conducive to the rapid diagnosis of patients. Combining the two methods, RT-PCR is used to inoculate cells for 2019-nCoV samples, which can reduce the cost of 2019-nCoV isolation by cell culture and save detection time, so that the analysis of biological particles in the isolated ambient air is sensitive, accurate, and comprehensive enough to detect the disinfection effect in the isolated environment.

### 2.6. Design of Isolated Diagnosis and Treatment Methods for the System

Mobile medical treatment is a way of treating patients through mobile communication technology, which is beneficial for COVID-19 isolation strategies. The use of this method allows clinicians to detect diseases anywhere and reduces the need for patients to travel to hospitals or clinics for diagnosis. Mobile diagnosis and treatment equipment is configured within a designated area of the isolation system, such as a RVs, vans, etc. This method may be used for 2019-nCoV detection, diagnosis, and treatment in the isolated area, thus saving isolation resources as well as diagnosis and treatment time.

#### Equipment for the Isolated Diagnosis and Treatment Facilities in the System

Mobile medical treatment utilizes a 5G telemedicine trolley. This equipment supports mobile nursing and mobile ward rounds, allows for remote physician consultations and orders, and promotes advertising and education [18]. The 5G telemedicine trolley can be used to perform mobile consultations through a remote audio and video interface anytime and anywhere. In addition, mobile ultrasound machines, defibrillation monitors, and other devices can be placed on 5G telemedicine trolleys to collect data, including vital signs, during the remote physical examination. Experts and frontline medical staff can use the 5G mobile medical vehicle to perform real-time interactive diagnosis and treatment concurrently, which greatly improves the transmission speed of medical diagnosis information, gives isolated patients more access to medical care, and to a certain extent, reduces workload during frontline medical staff deployment [19].

For in-vehicle medical facilities, the portability, operability, rapidity, and versatility of the equipment must be taken into consideration. The in-vehicle medical facilities must be equipped with a portable gas chromatography/mass spectrometry (GC/MS), quantitative real-time PCR, portable small biosafety cabinet, a fully automatic function, rapid detection system for pathogenic microorganism, mobile air sterilizer, microorganism sampling box, personal protective equipment, and epidemic prevention advertisement reports. Using the professional disposal platform, the health emergency team can not only rapidly carry out on-site work, such as disease monitoring and control, vector monitoring, environmental comprehensive treatment, personnel training, health education, and publicity, but also quickly test at least 100 indicators of infectious disease and drinking water samples [20].

The use of a mobile computerized tomography (CT) machine in the vehicle is a novel application of telemedicine [21]. The Internet-based, mobile medical mode of portable digital imaging equipment combines the digital imaging technology of CT with the transportation capability of an automobile and other technology such as the Internet.

## 3. Matters That Need Attention in System Development

Isolation and control of infectious sources and strengthening personal protection are the key measures for prevention and control. Once suspected cases are found, strict isolation measures should be taken to control the infectious sources and prevent the spread of the epidemic. Infectious diseases have the characteristics of an easy outbreak, rapid transmission and zoonosis, which will lead to relatively limited available infectious disease prevention and control resources to varying degrees. Therefore, a low-cost isolation tent that can transform an ordinary tent into an isolation tent, capable of receiving and treating infectious patients in a short time, can make up for this deficiency.

### 3.1. Treatment of Contaminants in the Isolation System

The medical and domestic waste generated in the isolation area must be treated according to medical waste disposal standards [22]. When the medical waste container is 75% full, the waste should be double-layer sealed, and before it leaves the isolation area, its surface should be sprayed with 1000 mg/L chlorine disinfectant, or a layer made from a medical waste packaging bag should be added. In addition, medical contaminants in the isolation area should be stored in a separate area from the living spaces, and the temporary storage place for medical contaminants must be disinfected with 1000 mg/L chlorine-containing disinfectant twice a day [23].

The contaminants produced by the isolated personnel must be promptly collected and disinfected by adding solid or liquid chlorine-containing disinfectant at a final concentration of 20 g/L effective chlorine, mixing and stirring evenly, and then allowing the disinfectant to act for more than 2 h. When disinfecting thick feces, the amount and time of disinfectant should be doubled. Disposable water-absorbing materials can be used to dip 5000–10,000 mg/L of chlorine-containing disinfectant and treat a small amount of contaminants. When disinfecting a large number of contaminants, the disinfectant powder or absorbent material containing water-absorbing ingredients must completely cover the contaminants, and then a sufficient amount of 5000–10,000 mg/L chlorine-containing disinfectant solution must be applied for more than 30 min to remove them [24,25].

### 3.2. Mental Health of Personnel in the Isolation System

#### 3.2.1. Causes of Mental Health Disorders in Isolated Patients

(1)The strictly enforced, 14-day period of medical isolation. People with suspected COVID-19 cannot leave the isolation area during this period, and all of their daily living must be carried out in this area only. Patients in isolation are not allowed to engage in outdoor activities, and their needs are limited, so they are prone to some psychological problems.(2)The isolation environment is frightening. People who are suspected of having COVID-19 must be isolated separately. The isolation environment may cause the isolated personnel to have a sense of inexplicable fear, which is aggravated by their unknown infection status.(3)Those who have been quarantined are eager to know their COVID-19 test results, receive medical diagnoses, and see imaging results immediately, but they are also worried about being diagnosed, affecting the lability of their mood [26,27].

#### 3.2.2. Measures to Address Psychological Distress in Isolated Personnel

The concept of “transparent isolation” is not new to modernist architecture, but this is the first time it has been applied to isolation vehicles. The isolation vehicle is a product in the field of mobile architecture. It has a moderate and compact size to dilute the existing accommodation density of the “shelter hospital” as little as possible and make it lighter. It is isolated, yet visible, and keeps those isolating away from each other so as to reduce the psychological pressure caused by several weeks of isolation. Its cost is so low that it can be washed and recycled, but it can also be destroyed after use, which is convenient for large-scale release. From a visual perspective, military and chemical areas, such as a CBRN negative pressure tent, should be avoided so as to provide a soft, light, transparent or translucent environment that can become part of the humanized medical space [28].

(1)The creation of a warm isolation environment may reduce anxiety in isolated patients. The staff in the isolation environment must communicate clearly while empathizing with the isolated patients [29]. Upon entering the isolation environment, staff must assuage patient concerns by familiarizing isolated patients with the equipment and functional facilities in the isolation environment. In addition, the relevant staff in the isolation environment must regularly communicate with the outside world, so that the quarantined staff are kept up-to-date with the prevention and control of COVID-19 and the latest developments of the epidemic [30]. At the same time, when isolated individuals have any problems, they should be able to communicate with the staff promptly.(2)The staff in the isolation area should evaluate the psychological condition of the isolated patients every day and with each change of shift. In addition, nurses specializing in psychological counseling must consult with isolated patients weekly. These consultations must include a careful assessment to determine any psychological needs; effective communication; knowledge assessments related to COVID-19; rational analyses related to fear in isolated patients; determination of appropriate response measures; and implementation of measures to reduce the fear in isolated patients due to lack of knowledge [31,32].

## 4. Conclusions

In this paper, a methodology for the development of a “private car” self-isolation ecosystem is proposed as a means of isolating patients with confirmed or suspected COVID-19 infection. Based on the understanding of the background, purpose, and significance of the system, the system has been designed by considering the following three aspects: system environment, disinfection, and diagnosis and treatment. Use of this system may reduce the burden caused by the exigent need for isolation resulting from the rapid increase of COVID-19 cases. It may also save isolation resources while providing a novel, adaptable method for patient isolation in infectious diseases in the future. In order to better deal with the spread of COVID-19 and major public health emergencies that may occur in the future, in addition to improving the rationality and scientificity of the allocation of isolation resources for epidemic prevention at ordinary times, more attention should be paid to the research and development of isolation facilities for epidemic prevention.

## Data Availability

Not applicable.

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
