# Peer review of "Conceptual Design of the “Private Car” Self-Isolation Ecosystem for the 2019-nCoV Infection"

_ijerph, 2022, doi:10.3390/ijerph191610385_

Round 1

Reviewer 1 Report

Thank you for giving me to review your manuscript. This manuscript is interesting for considering Self-Isolation Ecosystem for the COVID-19. However, the following points should be critical for the quality of the research.

The title should contain information on the research design “properly”

The abstract should contain the background information on this topic in international contexts.

The background should consist of theoretical framework, conceptual framework, and research question.

The background should focus on specific research questions interesting to international readers.

The authors should describe what kinds of methods of collecting the data as original article.

The authors should validate the number of participants.

The analysis section should be described deeply. What did the researchers inevestigate for development of self isolation ecosystem? This issue is critical.

Method and result parts should be described clearly as original article.

The discussion should summarize this research's results and significant points by revising the previous issues. 

Author Response

Dear Reviewer 1:

Thank you for your suggestions for modifying this article, which play an important role in improving the quality of this article. After careful consideration, the author of this article has carefully revised the paper, hoping to be approved. Revised portion are marked in red in the paper. The main corrections in the paper and the responds to the reviewer’s comments are as flowing:

  1. Response to comment: The title should contain information on the research design “properly”.

Response: We are very sorry that the title does not contain information on the reasearch design. Thanks for your valuable comments. We have changed the title to show the research design. The title is thus revised to present a conceptual design of the new system, which mignt be applied in the future.

  1. Response to comment: The abstract should contain the background information on this topic in international contexts.

Response: Thank you for your valuable suggestions. The abstract has presented “isolation of cases is one of the most effective methods for control and containment of COVID-19 and has been rapidly popularized”, in view of the current global world.

  1. Response to comment: The authors should describe what kinds of methods of collecting the data as original article.

Response: Thank you for your valuable comments. We have made correction according to the Reviewer’s comments. The specific question is to present “a self-isolation ecosystem of rapid deploying negative-pressurised “private car” is proposed for housing patients with 2019-nCoV infection, which could be lightweight, moderately sized, transparent to enable group supervision and communication”.

  1. Response to comment: The authors should validate the number of participants.

Response: Thank you for your valuable comments. As this is only the conceptual design, it has not been applied to the many participants, which would be planned in the long run.

  1. Response to comment: The analysis section should be described deeply. What did the researchers inevestigate for development of self isolation ecosystem? This issue is critical.

Response: Thank you for your valuable comments. The analysis is in part 2, which is detailed into six sections, i.e. design brief, Principles, the Isolation Environment, the Isolation Space, Isolation and Disinfection, Isolated Diagnosis and Treatment Methods.

  1. Response to comment: Method and result parts should be described clearly as original article.

Response: Considering the Reviewer’s suggestion, We have revised the discussion. The discussion is revised between 3. and 3.1.

  1. Response to comment: The discussion should summarize this research's results and significant points by revising the previous issues.

Response: Thank you for your valuable comments. We have made correction according to the Reviewer’s comments. The revision of the conclusion is also highlighted in the end.

Special thanks to you for your good comments.

Reviewer 2 Report

This paper describes the environmental, spatial, disinfection-related, diagnostic, and treatment design aspects of a proposed self-isolation ecosystem. Although it does not seem the most opportunistic moment in the timeline of COVID-19 pandemics, after the peak of the infectious waves, it may provide a method for patient isolation in infectious diseases in the future. The system has been designed by considering the system environment, disinfection, and diagnosis and treatment. In the equipment for the isolated diagnosis and treatment facilities, the authors propose the use of a mobile computerized tomography machine. In fact, a novel application of telemedicine. To combine the digital imaging technology with the transportation capability of an automobile and other technology such as the Internet does not seem to bear the proof of cost-benefit analysis and deserves further justification.

Author Response

Dear Reviewer 2:

Thank you for your suggestions for modifying this article, which play an important role in improving the quality of this article. After careful consideration, the author of this article has carefully revised the paper, hoping to be approved. Revised portion are marked in red in the paper. The main corrections in the paper and the responds to the reviewer’s comments are as flowing:

  1. Response to comment: To combine the digital imaging technology with the transportation capability of an automobile and other technology such as the Internet does not seem to bear the proof of cost-benefit analysis and deserves further justification.

Response: Thank you for your valuable comments. We have made correction according to the Reviewer’s comments. In view of the high cost and difficulty in popularization, we refer to some literatures and propose to establish a reasonable partition based on the current isolation tent (“private car”) and apply cheaper PVC and other materials. Moreover, our conceptual design also take the cost of applying digital method into consideration, and we instead propose the “transparent design” In the application.

Special thanks to you for your good comments.

Reviewer 3 Report

Thank you for the chance of reviewing the manuscript about developing “private car” self-isolation for the people infected with covid 19. The idea of this study is quite innovative, and the implication is highly significant. Therefore, I read it with great interest. However, there were some issues that needed to be addressed so as to implement a sustainable plan. The issues were mentioned as follows.

1.     For the introduction section, you may introduce some self- or non-self-isolation methods in different regions nowadays (remember to cite them) and then detailly indicate their limitations (e.g. lack of sufficient resources to support the whole isolation progress,…). After that, you may point out how important your strategy is in resolving the mentioned problems so that the urgency and significance of your work can be enhanced.

2.     For sections 2.2.1 Selection principles 2.2.1 and 2.2.2 Design of the isolated environment, you mentioned quite a lot of information. However, what are the rationales for suggesting principles and the ideas of the designs? You had better state the suggestion is based on what criteria (e.g. the standard of WHO, government, …)

3.     Regarding the mental health of personnel in the isolation system, you mentioned, “when isolated individuals have any problems, they should be able to communicate with the staff promptly” And “The staff in the isolation area should evaluate the psychological condition of the isolated patients every day and with each change of shift. In addition, nurses specializing in psychological counseling must consult with isolated patients weekly.” Even if the nursing staff responds to the patients quickly, their mental health might not be resolved. For example, the patients wanted to exercise so as to alleviate their mental health conditions. How can your strategy handle it? In fact, mental health problems may be one of the significant issues that the patients are concerned about the most. A better mental health condition may help improve the symptoms and side effects induced by the pandemic or isolation. However, your idea cannot be addressed them, and being isolated in a private car may limit their movement and they may not be able to exercise which may exacerbate their unhealthy physical and mental health conditions. Therefore, I suggest that you may think more about the mental issue to develop a more proper isolation environment and design for the patients. 

Author Response

Dear Reviewer 3:

Thank you for your suggestions for modifying this article, which play an important role in improving the quality of this article. After careful consideration, the author of this article has carefully revised the paper, hoping to be approved. Revised portion are marked in red in the paper. The main corrections in the paper and the responds to the reviewer’s comments are as flowing:

  1. Response to comment: For the introduction section, you may introduce some self- or non-self-isolation methods in different regions nowadays (remember to cite them) and then detailly indicate their limitations. After that, you may point out how important your strategy is in resolving the mentioned problems so that the urgency and significance of your work can be enhanced.

Response: Thank you for your valuable comments. We have made correction according to the Reviewer’s comments. The introduction is revised by providing the advantages and the disadvantages of isolation methods applied nowadays. Other detailed revision Is also presented.

  1. Response to comment: For sections 2.2.1 Selection principles 2.2.1 and 2.2.2 Design of the isolated environment, you mentioned quite a lot of information. However, what are the rationales for suggesting principles and the ideas of the designs? You had better state the suggestion is based on what criteria.

Response: Thank you for your valuable comments. For 2.2.1, the standards are deleted and substituted by the detailed design of negative pressure isolation unit, as the Isolation Environment of the System.

  1. Response to comment: I suggest that you may think more about the mental issue to develop a more proper isolation environment and design for the patients.

Response: We are very sorry for our negligence of mental problems in quarantined people. For the solution of mental problems encountered by the patients, a “transparent” design with novelty is described. Starting from the appearance design of the self isolation vehicle, we can adopt "transparent design" to make the patient feel "separated but not separated". This is the "transparent design concept" of architecture, and relevant documents have been applied in the design scheme of the field shelter hospital before.

Special thanks to you for your good comments.

Round 2

Reviewer 1 Report

The manuscript has been considerably improved. I think that this paper is suited for inclusion in our journal.

Reviewer 3 Report

Accepted